# AUXILIARY LEARNING BY IMPLICIT DIFFERENTIATION

**Aviv Navon**[*]
Bar-Ilan University, Israel
`aviv.navon@biu.ac.il`

**Idan Achituve**[*]
Bar-Ilan University, Israel
`idan.achituve@biu.ac.il`

**Haggai Maron**
NVIDIA, Israel
`hmaron@nvidia.com`

**Gal Chechik**[†]
Bar-Ilan University, Israel
NVIDIA, Israel
`gal.chechik@biu.ac.il`

**Ethan Fetaya**[†]
Bar-Ilan University, Israel
`ethan.fetaya@biu.ac.il`

## ABSTRACT

Training neural networks with auxiliary tasks is a common practice for improving the performance on a main task of interest. Two main challenges arise in this multi-task learning setting: (i) designing useful auxiliary tasks; and (ii) combining auxiliary tasks into a single coherent loss. Here, we propose a novel framework, *AuxiLearn*, that targets both challenges based on implicit differentiation. First, when useful auxiliaries are known, we propose learning a network that combines all losses into a single coherent objective function. This network can learn *non-linear* interactions between tasks. Second, when no useful auxiliary task is known, we describe how to learn a network that generates a meaningful, novel auxiliary task. We evaluate AuxiLearn in a series of tasks and domains, including image segmentation and learning with attributes in the low data regime, and find that it consistently outperforms competing methods.

## 1 INTRODUCTION

The performance of deep neural networks can significantly improve by training the main task of interest with additional auxiliary tasks (Goyal et al., 2019; Jaderberg et al., 2016; Mirowski, 2019). For example, learning to segment an image into objects can be more accurate when the model is simultaneously trained to predict other properties of the image like pixel depth or 3D structure (Standley et al., 2019). In the low data regime, models trained with the main task only are prone to overfit and generalize poorly to unseen data (Vinyals et al., 2016). In this case, the benefits of learning with multiple tasks are amplified (Zhang and Yang, 2017). Training with auxiliary tasks adds an inductive bias that pushes learned models to capture meaningful representations and avoid overfitting to spurious correlations.

In some domains, it may be easy to design beneficial auxiliary tasks and collect supervised data. For example, numerous tasks were proposed for self-supervised learning in image classification, including masking (Doersch et al., 2015), rotation (Gidaris et al., 2018) and patch shuffling (Doersch and Zisserman, 2017; Noroozi and Favaro, 2016). In these cases, it is not clear what would be the best way to combine all auxiliary tasks into a single loss (Doersch and Zisserman, 2017). The common practice is to compute a weighted combination of pretext losses by tuning the weights of individual losses using hyperparameter grid search. This approach, however, limits the potential of learning with auxiliary tasks because the run time of grid search grows exponentially with the number of tasks.

In other domains, obtaining good auxiliaries in the first place may be challenging or may require expert knowledge. For example, for point cloud classification, few self-supervised tasks have been proposed; however, their benefits so far are limited (Achituve et al., 2020; Hassani and Haley, 2019;

---

[*]Equal contributor
[†]Equal contributor

Sauder and Sievers, 2019; Tang et al., 2020). For these cases, it would be beneficial to automate the process of generating auxiliary tasks without domain expertise.

Our work takes a step forward in automating the use and design of auxiliary learning tasks. We name our approach *AuxiLearn*. AuxiLearn leverages recent progress made in implicit differentiation for optimizing hyperparameters (Liao et al., 2018; Lorraine et al., 2020). We demonstrate the effectiveness of AuxiLearn in two types of problems. First, in **combining auxiliaries**, for cases where auxiliary tasks are predefined. We describe how to train a deep neural network (NN) on top of auxiliary losses and combine them non-linearly into a unified loss. For instance, we combine per-pixel losses in image segmentation tasks using a convolutional NN (CNN). Second, **designing auxiliaries**, for cases where predefined auxiliary tasks are not available. We present an approach for learning such auxiliary tasks without domain knowledge and from input data alone. This is achieved by training an auxiliary network to generate auxiliary labels while training another, primary network to learn both the original task and the auxiliary task. One important distinction from previous works, such as (Kendall et al., 2018; Liu et al., 2019a), is that we do not optimize the auxiliary parameters using the training loss but rather on a separate (small) *auxiliary set*, allocated from the training data. This is a key difference since the goal of auxiliary learning is to improve generalization rather than help optimization on the training data.

To validate our proposed solution, we extensively evaluate AuxiLearn in several tasks in the low-data regime. In this regime, the models suffer from severe overfitting and auxiliary learning can provide the largest benefits. Our results demonstrate that using AuxiLearn leads to improved loss functions and auxiliary tasks, in terms of the performance of the resulting model on the main task. We complement our experimental section with two interesting theoretical insights regarding our model. The first shows that a relatively simple auxiliary hypothesis class may overfit. The second aims to understand which auxiliaries benefit the main task.

To summarize, we propose a novel general approach for learning with auxiliaries using implicit differentiation. We make the following novel contributions: (a) We describe a unified approach for combining multiple loss terms and for learning novel auxiliary tasks from the data alone; (b) We provide a theoretical observation on the capacity of auxiliary learning; (c) We show that the key quantity for determining beneficial auxiliaries is the Newton update; (d) We provide new results on a variety of auxiliary learning tasks with a focus on the low data regime. We conclude that implicit differentiation can play a significant role in automating the design of auxiliary learning setups.

## 2 RELATED WORK

**Learning with multiple tasks.** Multitask Learning (MTL) aims at simultaneously solving multiple learning problems while sharing information across tasks. In some cases, MTL benefits the optimization process and improves task-specific generalization performance compared to single-task learning (Standley et al., 2019). In contrast to MTL, auxiliary learning aims at solving a single, main task, and the purpose of all other tasks is to facilitate the learning of the primary task. At test time, only the main task is considered. This approach has been successfully applied in multiple domains, including computer vision (Zhang et al., 2014), natural language processing (Fan et al., 2017; Trinh et al., 2018), and reinforcement learning (Jaderberg et al., 2016; Lin et al., 2019).

**Dynamic task weighting.** When learning a set of tasks, the task-specific losses are combined into an overall loss. The way individual losses are combined is crucial because MTL-based models are sensitive to the relative weightings of the tasks (Kendall et al., 2018). A common approach for combining task losses is in a linear fashion. When the number of tasks is small, task weights are commonly tuned with a simple grid search. However, this approach does not extend to a large number of tasks, or a more complex weighting scheme. Several recent studies proposed scaling task weights using gradient magnitude (Chen et al., 2018), task uncertainty (Kendall et al., 2018), or the rate of loss change (Liu et al., 2019b). Sener and Koltun (2018) proposed casting the multitask learning problem as a multi-objective optimization. These methods assume that all tasks are equally important, and are less suited for auxiliary learning. Du et al. (2018) and Lin et al. (2019) proposed to weight auxiliary losses using gradient similarity. However, these methods do not scale well with the number of auxiliaries and do not take into account interactions between auxiliaries. In contrast, we propose to learn from data how to combine auxiliaries, possibly in a non-linear manner.

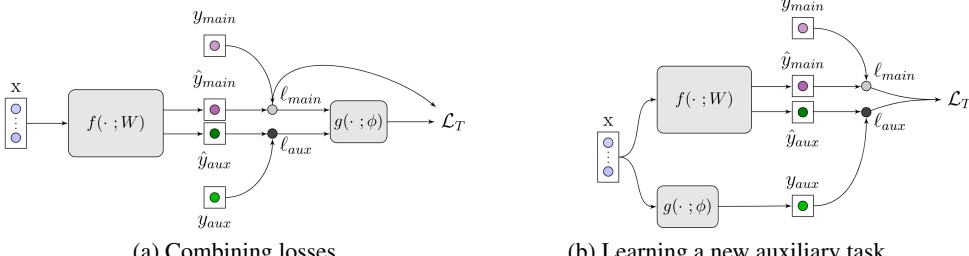

(a) Combining losses (b) Learning a new auxiliary task

Figure 1: The AuxiLearn framework. **(a)** Learning to combine losses into a single coherent loss term. Here, the auxiliary network operates over a vector of losses. **(b)** Generating a novel auxiliary task. Here the auxiliary network operates over the input space. In both cases, $g(\,\cdot\,;\phi)$ is optimized using IFT based on $\mathcal{L}_A$.

**Devising auxiliaries.** Designing an auxiliary task for a given main task is challenging because it may require domain expertise and additional labeling effort. For self-supervised learning (SSL), many approaches have been proposed (see Jing and Tian (2020) for a recent survey), but the joint representation learned through SSL may suffer from negative transfer and hurt the main task (Standley et al., 2019). Liu et al. (2019a) proposed learning a helpful auxiliary in a meta-learning fashion, removing the need for handcrafted auxiliaries. However, their system is optimized for the training data, which may lead to degenerate auxiliaries. To address this issue, an entropy term is introduced to force the auxiliary network to spread the probability mass across classes.

**Implicit differentiation based optimization.** Our formulation gives rise to a bi-level optimization problem. Such problems naturally arise in the context of meta-learning (Finn et al., 2017; Rajeswaran et al., 2019) and hyperparameter optimization (Bengio, 2000; Foo et al., 2008; Larsen et al., 1996; Liao et al., 2018; Lorraine et al., 2020; Pedregosa, 2016). The Implicit Function Theorem (IFT) is often used for computing gradients of the upper-level function, this operation requires calculating a vector-inverse Hessian product. However, for modern neural networks, it is infeasible to calculate it explicitly, and an approximation must be devised. Luketina et al. (2016) proposed approximating the Hessian with the identity matrix, whereas Foo et al. (2008); Pedregosa (2016); Rajeswaran et al. (2019) used conjugate gradient (CG) to approximate the product. Following Liao et al. (2018); Lorraine et al. (2020), we use a truncated Neumann series and efficient vector-Jacobian products, as it was empirically shown to be more stable than CG.

## 3 OUR METHOD

We now describe the general AuxiLearn framework for learning with auxiliary tasks. For that purpose, we use two networks, a primary network that is optimized on all tasks and an auxiliary network that is optimized on the main task only. First, we introduce our notations and formulate the general objective. Then, we describe two instances of this framework: combining auxiliaries and learning new auxiliaries. Finally, we present our optimization approach for both instances.

### 3.1 PROBLEM DEFINITION

Let $\{(\mathbf{x}_i^t, \mathbf{y}_i^t)\}_i$ be the training set and $\{(\mathbf{x}_i^a, \mathbf{y}_i^a)\}_i$ be a distinct independent set which we term *auxiliary set*. Let $f(\,\cdot\,;W)$ denote the primary network, and let $g(\,\cdot\,;\phi)$ denote the auxiliary network. Here, $W$ are the parameters of the model optimized on the training set, and $\phi$ are the auxiliary parameters trained on the auxiliary set. The training loss is defined as:

$$\mathcal{L}_T = \mathcal{L}_T(W, \phi) = \sum_i \ell_{main}(\mathbf{x}_i^t, \mathbf{y}_i^t; W) + h(\mathbf{x}_i^t, \mathbf{y}_i^t, W; \phi), \tag{1}$$

where $\ell_{main}$ denotes the loss of the main task and $h$ is the overall auxiliary loss, controlled by $\phi$. In Sections 3.2 & 3.3 we will describe two instances of $h$. We note that $h$ has access to both $W$ and $\phi$. The loss on the auxiliary set is defined as $\mathcal{L}_A = \sum_i \ell_{main}(\mathbf{x}_i^a, \mathbf{y}_i^a; W)$, since we are interested in the generalization performance of the main task.

We wish to find auxiliary parameters ($\phi$) such that the primary parameters ($W$), trained with the combined objective, generalize well. More formally, we seek

$$\phi^* = \arg\min_{\phi} \mathcal{L}_A(W^*(\phi)), \quad \text{s.t.} \quad W^*(\phi) = \arg\min_W \mathcal{L}_T(W, \phi). \tag{2}$$

## 3.2 Learning to combine auxiliary tasks

Suppose we are given $K$ auxiliary tasks, usually designed using expert domain knowledge. We wish to learn how to optimally leverage these auxiliaries by learning to combine their corresponding losses. Let $\boldsymbol{\ell}(\mathbf{x}, \boldsymbol{y}; W) = (\ell_{main}(\mathbf{x}, y^{main}; W), \ell_1(\mathbf{x}, y^1; W), ..., \ell_K(\mathbf{x}, y^K; W))$ denote a loss vector. We wish to learn an auxiliary network $g : \mathbb{R}^{K+1} \to \mathbb{R}$ over the losses that will be added to $\ell_{main}$ in order to output the training loss $\mathcal{L}_T = \ell_{main} + g(\boldsymbol{\ell}; \phi)$. Here, $h$ from Eq. (1) is given by $h(\cdot\,; \phi) = g(\boldsymbol{\ell}; \phi)$.

Typically, $g(\boldsymbol{\ell}; \phi)$ is chosen to be a linear combination of the losses: $g(\boldsymbol{\ell}; \phi) = \sum_j \phi_j \ell_j$, with positive weights $\phi_j \geq 0$ that are tuned using a grid search. However, this method can only scale to a few auxiliaries, as the run time of grid search is exponential in the number of tasks. Our method can handle a large number of auxiliaries and easily extends to a more flexible formulation in which $g$ parametrized by a deep NN. This general form allows us to capture complex interactions between tasks, and learn non-linear combinations of losses. See Figure 1a for illustration.

One way to view a non-linear combination of losses is as an adaptive linear weighting, where losses have a different set of weights for each datum. If the loss at point $\mathbf{x}$ is $\ell_{main}(\mathbf{x}, y^{main}) + g(\boldsymbol{\ell}(\mathbf{x}, \boldsymbol{y}))$, then the gradients are $\nabla_W \ell_{main}(\mathbf{x}, y^{main}) + \sum_j \frac{\partial g}{\partial \ell_j} \nabla_W \ell_j(\mathbf{x}, y^j)$. This is equivalent to an adaptive loss where the loss of datum $\mathbf{x}$ is $\ell_{main} + \sum_j \alpha_{j,\mathbf{x}} \ell_j$ and, $\alpha_{j,\mathbf{x}} = \frac{\partial g}{\partial \ell_j}$. This observation connects our approach to other studies that assign adaptive loss weighs (e.g., Du et al. (2018); Liu et al. (2019b)).

**Convolutional loss network.** In certain problems there exists a spatial relation among losses. For example, semantic segmentation and depth estimation for images. A common approach is to average the losses over all locations. In contrast, AuxiLearn can leverage this spatial relation for creating a *loss-image* in which each task forms a channel of pixel-losses induced by the task. We then parametrize $g$ as a CNN that acts on this loss-image. This yields a spatial-aware loss function that captures interactions between task losses. See an example of a loss image in Figure 3

**Monotonicity.** It is common to parametrize the function $g(\boldsymbol{\ell}; \phi)$ as a linear combination with non-negative weights. Under this parameterization, $g$ is a monotonic non-decreasing function of the losses. A natural question that arises is whether we should generalize this behavior and constrain $g(\boldsymbol{\ell}; \phi)$ to be non-decreasing w.r.t. the input losses as well? Empirically, we found that training with monotonic non-decreasing networks tends to be more stable and has a better or equivalent performance. We impose monotonicity during training with negative weights clipping. See Appendix C.2 for a detailed discussion and empirical comparison to non-monotonic networks.

## 3.3 Learning new auxiliary tasks

The previous subsection focused on situations where auxiliary tasks are given. In many cases, however, no useful auxiliary tasks are known in advance, and we are only presented with the main task. We now describe how to use AuxiLearn in such cases. The intuition is simple: We wish to learn an auxiliary task that pushes the representation of the primary network to generalize better on the main task, as measured using the auxiliary set. We do so in a student-teacher manner: an auxiliary "teacher" network produces labels for the primary network (the "student") which tries to predict these labels as an auxiliary task. Both networks are trained jointly.

More specifically, for auxiliary classification, we learn a soft labeling function $g(\mathbf{x}; \phi)$ which produces pseudo labels $y_{aux}$ for input samples $\mathbf{x}$. These labels are then provided to the main network $f(\mathbf{x}; W)$ for training (see Figure 1b). During training, the primary network $f(\mathbf{x}; W)$ outputs two predictions, $\hat{y}_{main}$ for the main task and $\hat{y}_{aux}$ for the auxiliary task. We then compute the full training loss $\mathcal{L}_T = \ell_{main}(\hat{y}_{main}, y_{main}) + \ell_{aux}(\hat{y}_{aux}, y_{aux})$ to update $W$. Here, the $h$ component of $\mathcal{L}_T$ in Eq. (1) is given by $h(\cdot\,; \phi) = \ell_{aux}(f(\mathbf{x}_i^t; W), g(\mathbf{x}_i^t; \phi))$. As before, we update $\phi$ using the auxiliary set with the loss $\mathcal{L}_A = \ell_{main}$. Intuitively, the teacher auxiliary network $g$ is rewarded when it provides labels to the student that help it succeed in the main task, as measured using $\mathcal{L}_A$.

### 3.4 OPTIMIZING AUXILIARY PARAMETERS

We now return to the bi-level optimization problem in Eq. (2) and present the optimizing method for $\phi$. Solving Eq. (2) for $\phi$ poses a problem due to the indirect dependence of $\mathcal{L}_A$ on the auxiliary parameters. To compute the gradients of $\phi$, we need to differentiate through the optimization process over $W$, since $\nabla_\phi \mathcal{L}_A = \nabla_W \mathcal{L}_A \cdot \nabla_\phi W^*$. As in Liao et al. (2018); Lorraine et al. (2020), we use the implicit function theorem (IFT) to evaluate $\nabla_\phi W^*$:

$$\nabla_\phi W^* = - \underbrace{(\nabla_W^2 \mathcal{L}_T)^{-1}}_{|W| \times |W|} \cdot \underbrace{\nabla_\phi \nabla_W \mathcal{L}_T}_{|W| \times |\phi|}. \tag{3}$$

We can leverage the IFT to approximate the gradients of the auxiliary parameters $\phi$:

$$\nabla_\phi \mathcal{L}_A(W^*(\phi)) = - \underbrace{\nabla_W \mathcal{L}_A}_{1 \times |W|} \cdot \underbrace{(\nabla_W^2 \mathcal{L}_T)^{-1}}_{|W| \times |W|} \cdot \underbrace{\nabla_\phi \nabla_W \mathcal{L}_T}_{|W| \times |\phi|}. \tag{4}$$

See Appendix A for a detailed derivation. To compute the vector and Hessian inverse product, we use the algorithm proposed by Lorraine et al. (2020), which uses Neumann approximation and efficient vector-Jacobian product. We note that accurately computing $\nabla_\phi \mathcal{L}_A$ by IFT requires finding a point such that $\nabla_W \mathcal{L}_T = 0$. In practice, we only approximate $W^*$, and simultaneously train both $W$ and $\phi$ by altering between optimizing $W$ on $\mathcal{L}_T$, and optimizing $\phi$ using $\mathcal{L}_A$. We summarize our method in Alg. 1 and 2. Theoretical considerations regarding our method are given in Appendix D.

---

**Algorithm 1:** AuxiLearn

Initialize auxiliary parameters $\phi$ and weights $W$;
  **while** *not converged* **do**
    **for** $k = 1, ..., N$ **do**
      $\mathcal{L}_T = \ell_{main}(\mathbf{x}, y; W) + h(\mathbf{x}, y, W; \phi)$
      $W \leftarrow W - \alpha \nabla_W \mathcal{L}_T \big|_{\phi, W}$
    **end**
    $\phi \leftarrow \phi - \text{Hypergradient}(\mathcal{L}_A, \mathcal{L}_T, \phi, W)$
**end**
**return** $W$

---

**Algorithm 2:** Hypergradient

**Input:** training loss $\mathcal{L}_T$, auxiliary loss $\mathcal{L}_A$, a
       fixed point $(\phi', W^*)$, number of
       iterations $J$, learning rate $\alpha$
$v = p = \nabla_W \mathcal{L}_A |_{(\phi', W^*)}$
**for** $j = 1, ..., J$ **do**
    $v \mathrel{-}= \alpha v \cdot \nabla_W \nabla_W \mathcal{L}_T$
    $p \mathrel{+}= v$
**end**
**return** $-p \nabla_\phi \nabla_W \mathcal{L}_T |_{(\phi', W^*)}$

---

## 4 ANALYSIS

### 4.1 COMPLEXITY OF AUXILIARY HYPOTHESIS SPACE

In our learning setup, an additional auxiliary set is used for tuning a large set of auxiliary parameters. A natural question arises: could the auxiliary parameters overfit this auxiliary set? and what is the complexity of the auxiliary hypothesis space $\mathcal{H}_\phi$? Analyzing the complexity of this space is difficult because it is coupled with the hypothesis space $\mathcal{H}_W$ of the main model. One can think of this hypothesis space as a subset of the original model hypothesis space $\mathcal{H}_\phi = \{h_W : \exists \phi \text{ s.t. } W = \arg\min_W \mathcal{L}_T(W, \phi)\} \subset \mathcal{H}_W$. Due to the coupling with $\mathcal{H}_W$ the behavior can be unintuitive. We show that even simple auxiliaries can have infinite VC dimensions.

**Example:** Consider the following 1D hypothesis space for binary classification $\mathcal{H}_W = \{\lceil \cos(Wx) \rceil, W \in \mathbb{R}\}$, which has infinite VC-dimension. Let the main loss be the zero-one loss and the auxiliary loss be $h(\phi, W) = (\phi - W)^2$, namely, an $L_2$ regularization with a learned center. Since the model hypothesis space $\mathcal{H}_W$ has an infinite VC-dimension, there exist training and auxiliary sets of any size that are shattered by $\mathcal{H}_W$. Therefore, for any labeling of the auxiliary and training sets, we can let $\phi = \hat{\phi}$, the parameter that perfectly classifies both sets. We then have that $\hat{\phi}$ is the optimum of the training with this auxiliary loss and we get that $\mathcal{H}_\phi$ also has an infinite VC-dimension.

This important example shows that even seemingly simple-looking auxiliary losses can overfit due to the interaction with the model hypothesis space. Thus, it motivates our use of a separate auxiliary set.

### 4.2 ANALYZING AN AUXILIARY TASK EFFECT

When designing or learning auxiliary tasks, one important question is, what makes an auxiliary task useful? Consider the following loss with a single auxiliary task $\mathcal{L}_T(W, \phi) = \sum_i \ell_{main}(\mathbf{x}_i^t, \boldsymbol{y}_i^t, W) +$

$\phi \cdot \ell_{aux}(\mathbf{x}_i^t, \boldsymbol{y}_i^t, W)$. Here $h = \phi \cdot \ell_{aux}$. Assume $\phi = 0$ so we optimize $W$ only on the standard main task loss. We can now check if $\frac{d\mathcal{L}_A}{d\phi}|_{\phi=0} > 0$, namely would it help to add this auxiliary task?

**Proposition 1.** *Let* $\mathcal{L}_T(W, \phi) = \sum_i \ell_{main}(\mathbf{x}_i^t, \boldsymbol{y}_i^t, W) + \phi \cdot \ell_{aux}(\mathbf{x}_i^t, \boldsymbol{y}_i^t, W)$. *Suppose that* $\phi = 0$ *and that the main task was trained until convergence. We have*

$$\left.\frac{d\mathcal{L}_A(W^*(\phi))}{d\phi}\right|_{\phi=0} = -\langle \nabla_W \mathcal{L}_A^T, \nabla_W^2 \mathcal{L}_T^{-1} \nabla_W \mathcal{L}_T \rangle, \tag{5}$$

*i.e. the gradient with respect to the auxiliary weight is the inner product between the Newton methods update and the gradient of the loss on the auxiliary set.*

*Proof.* In the general case, the following holds $\frac{d\mathcal{L}_A}{d\phi} = -\nabla_W \mathcal{L}_A (\nabla_W^2 \mathcal{L}_T)^{-1} \nabla_\phi \nabla_W \mathcal{L}_T$. For a linear combination, we have $\nabla_\phi \nabla_W \mathcal{L}_T = \sum_i \nabla_W \ell_{aux}(\mathbf{x}_i^t, \boldsymbol{y}_i^t)$. Since $W$ is optimized till convergence of the main task we obtain $\nabla_\phi \nabla_W \mathcal{L}_T = \nabla_W \mathcal{L}_T$. □

This simple result shows that the key quantity to observe is the Newton update, rather than the gradient which is often used (Lin et al., 2019; Du et al., 2018). Intuitively, the Newton update is the important quantity because if $\Delta\phi$ is small then we are almost at the optimum. Then, due to quadratic convergence, a single Newton step is sufficient for approximately converging to the new optimum.

## 5 EXPERIMENTS

We evaluate the AuxiLearn framework in a series of tasks of two types: combining given auxiliary tasks into a unified loss (Sections 5.1 - 5.3), and generating a new auxiliary task (Section 5.4). Further experiments and analysis of both modules are given in Appendix C. Throughout all experiments, we use an extra data split for the auxiliary set. Hence, we use four data sets: training set, validation set, test set, and auxiliary set. The samples for the auxiliary set are pre-allocated from the training set. For a fair comparison, these samples are used as part of the training set by all competing methods. Effectively, this means we have a slightly smaller training set for optimizing the parameters $W$ of the primary network. In all experiments, we report the mean performance (e.g., accuracy) along with the Standard Error of the Mean (SEM). Full implementation details of all experiments are given in Appendix B. Our code is available at https://github.com/AvivNavon/AuxiLearn.

**Model variants.** For learning to combine losses, we evaluated the following variants of auxiliary networks: **(1) Linear**: A convex linear combination between the loss terms; **(2) Linear neural network (Deep linear)**: A deep fully-connected NN with linear activations; **(3) Nonlinear**: A standard feed-forward NN over the loss terms. For Section 5.3 only **(4) ConvNet**: A CNN over the loss-images. The expressive power of the deep linear network is equivalent to that of a 1-layer linear network; However, from an optimization perspective, it was shown that the over-parameterization introduced by the network's depth could stabilize and accelerate convergence (Arora et al., 2018; Saxe et al., 2014). All variants are constrained to represent only monotone non-decreasing functions.

### 5.1 AN ILLUSTRATIVE EXAMPLE

We first present an illustrative example of how AuxiLearn changes the loss landscape and helps generalization in the presence of label noise and harmful tasks. Consider a regression problem with $y_{main} = \mathbf{w}^{\star T}\mathbf{x} + \epsilon_0$ and two auxiliary tasks. The first auxiliary is helpful, $y_1 = \mathbf{w}^{\star T}\mathbf{x} + \epsilon_1$, whereas the second auxiliary is harmful $y_2 = \tilde{\mathbf{w}}^T\mathbf{x} + \epsilon_2$, $\tilde{\mathbf{w}} \neq \mathbf{w}^\star$. We let

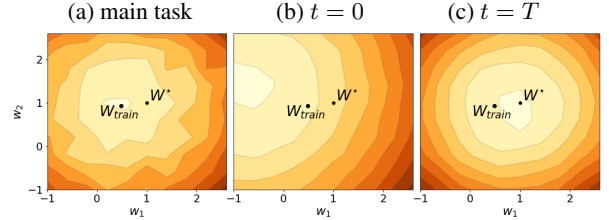

Figure 2: Loss landscape generated by the auxiliary network. Darker is higher. See text for details.

$\epsilon_0 \sim \mathcal{N}(0, \sigma_{main}^2)$ and $\epsilon_1, \epsilon_2 \sim \mathcal{N}(0, \sigma_{aux}^2)$, with $\sigma_{main}^2 > \sigma_{aux}^2$. We optimize a linear model with weights $\mathbf{w} \in \mathbb{R}^2$ that are shared across tasks, i.e., no task-specific parameters. We set $\mathbf{w}^\star = (1, 1)^T$ and $\tilde{\mathbf{w}} = (2, -4)^T$. We train an auxiliary network to output linear task weights and observe the changes to the loss landscape in Figure 2. The left plot shows the loss landscape for the main task,

| (a) image | (b) GT labels | (c) aux. loss | (d) main loss | (e) pix. weight |
|---|---|---|---|---|

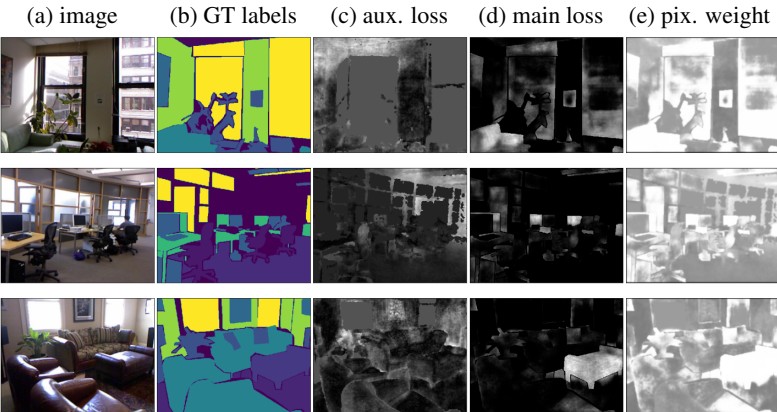

Figure 3: *Loss images* on test examples from NYUv2: **(a)** original image; **(b)** semantic segmentation ground truth; **(c)** auxiliaries loss; **(d)** segmentation (main task) loss; **(e)** adaptive pixel-wise weight $\sum_j \partial \mathcal{L}_T / \partial \ell_j$.

with a training set optimal solution $\mathbf{w}_{train}$. Note that $\mathbf{w}_{train} \neq \mathbf{w}^*$ due to the noise in the training data. The loss landscape of the weighted train loss at the beginning ($t = 0$) and the end ($t = T$) of training is shown in the middle and right plots, respectively. Note how AuxiLearn learns to ignore the harmful auxiliary and use the helpful one to find a better solution by changing the loss landscape. In Appendix C.3 we show that the auxiliary task weight is inversely proportional to the label noise.

## 5.2 FINE-GRAINED CLASSIFICATION WITH MANY AUXILIARY TASKS

In fine-grained visual classification tasks, annotators should have domain expertise, making data labeling challenging and potentially expensive (e.g., in the medical domain). In some cases, however, non-experts can annotate visual attributes that are informative about the main task. As an example, consider the case of recognizing bird species, which would require an ornithologist, yet a layman can describe the head color or bill shape of a bird. These features naturally form auxiliary tasks, which can be leveraged for training jointly with the main task of bird classification.

We evaluate AuxiLearn in this setup of fine-grained classification using the Caltech-UCSD Birds 200-2011 dataset (CUB) (Wah et al., 2011). CUB contains 200 bird species in 11,788 images, each associated with a set of 312 binary visual attributes, which we use as auxiliaries. Since we are interested in setups where optimizing the main task alone does not generalize well, we demonstrate our method in a semi-supervised setting: we assume that auxiliary labels are available for all images but only $K$ labels per class are available for the main task (noted as $K$-shot).

Table 1: Test classification accuracy on CUB 200-2011 dataset, averaged over three runs ($\pm$ SEM).

| | 5-shot | | 10-shot | |
|---|---|---|---|---|
| | Top 1 | Top 3 | Top 1 | Top 3 |
| STL | $35.50 \pm 0.7$ | $54.79 \pm 0.7$ | $54.79 \pm 0.3$ | $74.00 \pm 0.1$ |
| Equal | $41.47 \pm 0.4$ | $62.62 \pm 0.4$ | $55.36 \pm 0.3$ | $75.51 \pm 0.4$ |
| Uncertainty | $35.22 \pm 0.3$ | $54.99 \pm 0.7$ | $53.75 \pm 0.6$ | $73.25 \pm 0.3$ |
| DWA | $41.82 \pm 0.1$ | $62.91 \pm 0.4$ | $54.90 \pm 0.3$ | $75.74 \pm 0.3$ |
| GradNorm | $41.49 \pm 0.4$ | $63.12 \pm 0.4$ | $55.23 \pm 0.1$ | $75.62 \pm 0.3$ |
| GCS | $42.57 \pm 0.7$ | $62.60 \pm 0.1$ | $55.65 \pm 0.2$ | $75.71 \pm 0.1$ |
| **AuxiLearn** | | | | |
| Linear | $41.71 \pm 0.4$ | $63.73 \pm 0.6$ | $54.77 \pm 0.2$ | $75.51 \pm 0.7$ |
| Deep Linear | $45.84 \pm 0.3$ | $66.21 \pm 0.5$ | $57.08 \pm 0.2$ | $75.3 \pm 0.6$ |
| Nonlinear | $\mathbf{47.07 \pm 0.1}$ | $\mathbf{68.25 \pm 0.3}$ | $\mathbf{59.04 \pm 0.2}$ | $\mathbf{78.08 \pm 0.2}$ |

We compare AuxiLearn with the following MTL and auxiliary learning baselines: **(1) Single-task learning (STL):** Training only on the main task. **(2) Equal:** Standard multitask learning with equal weights for all auxiliary tasks. **(3) GradNorm** (Chen et al., 2018): An MTL method that scales losses based on gradient magnitude. **(4) Uncertainty** (Kendall et al., 2018): An MTL approach that uses task uncertainty to adjust task weights. **(5) Gradient Cosine Similarity (GCS)** (Du et al., 2018): An auxiliary-learning approach that uses gradient similarity between the main and auxiliary tasks. **(6) Dynamic weight averaging (DWA)** (Liu et al., 2019b): An MTL approach that sets task weights based on the rate of loss change over time. The primary network in all experiments is ResNet-18 (He et al., 2016) pre-trained on ImageNet. We use a 5-layer fully connected NN for the auxiliary network. Sensitivity analysis of the network size and auxiliary set size is presented in Appendix C.4.

Table 1 shows the test set classification accuracy. Most methods significantly improve over the STL baseline, highlighting the benefits of using additional (weak) labels. Our *Nonlinear* and *Deep linear* auxiliary network variants outperform all previous approaches by a large margin. As expected, a non-linear auxiliary network is better than its linear counterparts. This suggests that there are some non-linear interactions between the loss terms that the non-linear network is able to capture. Also, notice the effect of using deep-linear compared to a (shallow) linear model. This result indicates that at least part of the improvement achieved by our method is attributed to the over-parameterization of the auxiliary network. In the Appendix we further analyze properties of auxiliary networks. Appendix C.5 visualizes the full optimization path of a linear auxiliary network over a polynomial kernel on the losses, and Appendix C.6 shows that the last state of the auxiliary network is not informative enough.

## 5.3 PIXEL-WISE LOSSES

We consider the indoor-scene segmentation task from Couprie et al. (2013), that uses the NYUv2 dataset (Silberman et al., 2012). We consider the 13-class semantic segmentation as the main task, with depth and surface-normal prediction (Eigen and Fergus, 2015) as auxiliaries. We use SegNet (Badrinarayanan et al., 2017) based model for the primary network, and a 4-layer CNN for the auxiliary network.

Since losses in this task are given per-pixel, we can apply the ConvNet variant of the auxiliary network to the loss image. Namely, each task forms a channel with the per-pixel losses as values. Table 2 reports the mean Intersection over Union (mIoU) and pixel accuracy for the main segmentation task. Here, we

Table 2: Test results for semantic segmentation on NYUv2, averaged over four runs ($\pm$ SEM).

|  | mIoU | Pixel acc. |
|---|---|---|
| STL | $18.90 \pm 0.21$ | $54.74 \pm 0.94$ |
| Equal | $19.20 \pm 0.19$ | $55.37 \pm 1.00$ |
| Uncertainty | $19.34 \pm 0.18$ | $55.70 \pm 0.79$ |
| DWA | $19.38 \pm 0.14$ | $55.37 \pm 0.35$ |
| GradNorm | $19.52 \pm 0.21$ | $56.70 \pm 0.33$ |
| MGDA | $19.53 \pm 0.35$ | $56.28 \pm 0.46$ |
| GCS | $19.94 \pm 0.13$ | $56.58 \pm 0.81$ |
| **AuxiLearn (ours)** | | |
| Linear | $20.04 \pm 0.38$ | $\mathbf{56.80 \pm 0.14}$ |
| Deep Linear | $19.94 \pm 0.12$ | $56.45 \pm 0.79$ |
| Nonlinear | $20.09 \pm 0.34$ | $\mathbf{56.80 \pm 0.53}$ |
| ConvNet | $\mathbf{20.54 \pm 0.30}$ | $56.69 \pm 0.44$ |

also compare with MGDA (Sener and Koltun, 2018) which had extremely long training time in CUB experiments due to the large number of auxiliary tasks, and therefore was not evaluated in Section 5.2. All weighting methods achieve a performance gain over the STL model. The ConvNet variant of AuxiLearn outperforms all competitors in terms of test mIoU.

Figure 3 shows examples of the loss-images for the auxiliary (c) and main (d) tasks, together with the pixel-wise weights (e). First, note how the loss-images resemble the actual input images. This suggests that a spatial relationship can be leveraged using a CNN auxiliary network. Second, pixel weights are a non-trivial combination of the main and auxiliary task losses. In the top (bottom) row, the plant (couch) has a low segmentation loss and intermediate auxiliary loss. As a result, a higher weight is allocated to these pixels which increases the error signal.

## 5.4 LEARNING AUXILIARY LABELS

Table 3: Learning auxiliary task. Test accuracy averaged over three runs ($\pm$SEM) without pre-training.

|  | CIFAR10 (5%) | CIFAR100 (5%) | SVHN (5%) | CUB (30-shot) | Pet (30-shot) | Cars (30-shot) |
|---|---|---|---|---|---|---|
| STL | $50.8 \pm 0.8$ | $19.8 \pm 0.7$ | $72.9 \pm 0.3$ | $37.2 \pm 0.8$ | $26.1 \pm 0.5$ | $59.2 \pm 0.4$ |
| MAXL-F | $56.1 \pm 0.1$ | $20.4 \pm 0.6$ | $75.4 \pm 0.3$ | $39.6 \pm 1.3$ | $26.2 \pm 0.3$ | $59.6 \pm 1.1$ |
| MAXL | $58.2 \pm 0.3$ | $21.0 \pm 0.4$ | $75.5 \pm 0.4$ | $40.7 \pm 0.6$ | $26.3 \pm 0.6$ | $60.4 \pm 0.8$ |
| **AuxiLearn** | $\mathbf{60.7 \pm 1.3}$ | $\mathbf{21.5 \pm 0.3}$ | $\mathbf{76.4 \pm 0.2}$ | $\mathbf{44.5 \pm 0.3}$ | $\mathbf{37.0 \pm 0.6}$ | $\mathbf{64.4 \pm 0.3}$ |

In many cases, designing helpful auxiliaries is challenging. We now evaluate AuxiLearn in learning multi-class classification auxiliary tasks. We use three multi-class classification datasets: CIFAR10, CIFAR100 (Krizhevsky et al., 2009), SVHN (Netzer et al., 2011), and three fine-grained classification datasets: CUB-200-2011, Oxford-IIIT Pet (Parkhi et al., 2012), and Cars (Krause et al., 2013). Pet contains 7349 images of 37 species of dogs and cats, and Cars contains 16,185 images of 196 cars.

Following Liu et al. (2019a), we learn a different auxiliary task for each class of the main task. In all experiments and all learned tasks, we set the number of classes to 5 . To examine the effect of the learned auxiliary losses in the low-data regime, we evaluate the performance while training with only 5% of the training set in CIFAR10, CIFAR100, and SVHN datasets, and $\sim$ 30 samples per

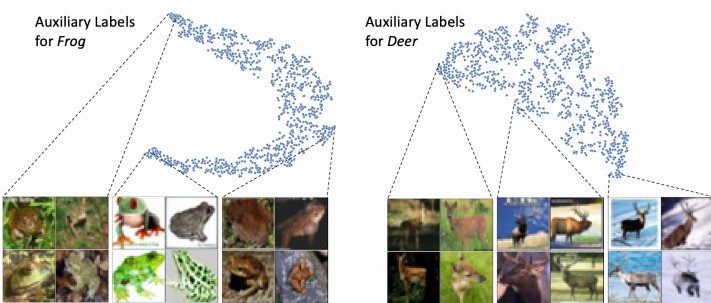

Figure 4: t-SNE applied to auxiliary labels learned for *Frog* and *Deer* classes, in CIFAR10. Best viewed in color.

class in CUB, Oxford-IIIT Pet, and Cars. We use VGG-16 (Simonyan and Zisserman, 2014) as the backbone for both CIFAR datasets, a 4-layers ConvNet for the SVHN experiment, and ResNet18 for the fine-grained datasets. In all experiments, the architectures of the auxiliary and primary networks were set the same and were trained from scratch without pre-training.

We compared our approach with the following baselines: **(1) Single-task learning (STL):** Training the main task only. **(2) MAXL:** Meta AuXiliary Learning (MAXL) proposed by Liu et al. (2019a) for learning auxiliary tasks. MAXL optimizes the label generator in a meta-learning fashion. **(3) MAXL-F:** A frozen MAXL label generator, that is initialized randomly. It decouples the effect of having a teacher network from the additional effect brought by the training process.

Table 3 shows that AuxiLearn outperforms all baselines in all setups, even-though it sacrifices some of the training set for the auxiliary set. It is also worth noting that our optimization approach is significantly faster than MAXL, yielding $\times 3$ improvement in run-time. In Appendix C.9 and C.10 we show additional experiments for this setup, including an extension of the method to point-cloud part segmentation and experiments with varying training data sizes.

Figure 4 presents a 2D t-SNE projection of the 5D vector of auxiliary (soft) labels that are learned using AuxiLearn. We use samples of the main classes *Frog* (left) and *Deer* (right) from the CIFAR10 dataset. t-SNE was applied to each auxiliary task separately. When considering how images are projected to this space of auxiliary soft labels, several structures emerge. The auxiliary network learns a fine partition of the *Frog* class that separates real images from illustrations. More interesting, the soft labels learned for the class *Deer* have a middle region that only contains deers with antlers (in various poses and varying backgrounds). By capturing this semantic feature in the learned auxiliary labels, the auxiliary task can help the primary network to discriminate between main task classes.

## 6 Discussion

In this paper, we presented a novel and unified approach for two tasks: combining predefined auxiliary tasks, and learning auxiliary tasks that are useful for the primary task. We theoretically showed which auxiliaries can be beneficial and the importance of using a separate auxiliary set. We empirically demonstrated that our method achieves significant improvement over existing methods on various datasets and tasks. This work opens interesting directions for future research. First, when training deep linear auxiliary networks, we observed similar learning dynamics to those of non-linear models. As a result, they generated better performance compared to their linear counterparts. This effect was observed in standard training setup, but the optimization path in auxiliary networks is very different. Second, we find that reallocating labeled data from the training set to an auxiliary set is consistently helpful. A broader question remains what is the most efficient allocation.

## Acknowledgements

This study was funded by a grant to GC from the Israel Science Foundation (ISF 737/2018), and by an equipment grant to GC and Bar-Ilan University from the Israel Science Foundation (ISF 2332/18). IA and AN were funded by a grant from the Israeli innovation authority, through the AVATAR consortium.

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

# Appendix: Auxiliary Learning by Implicit Differentiation

## A  GRADIENT DERIVATION

We provide here the derivation of Eq. (4) in Section 3. One can look at the function $\nabla_W \mathcal{L}_T(W, \phi)$ around a certain local-minima point $(\hat{W}, \hat{\phi})$ and assume the Hessian $\nabla_W^2 \mathcal{L}_T(\hat{W}, \hat{\phi})$ is positive-definite. At that point, we have $\nabla_W \mathcal{L}_T(\hat{W}, \hat{\phi}) = 0$. From the IFT, we have that locally around $(\hat{W}, \hat{\phi})$, there exists a smooth function $W^*(\phi)$ such that $\nabla_W \mathcal{L}_T(W, \phi) = 0$ if $W = W^*(\phi)$. Since the function $\nabla_W \mathcal{L}_T(W^*(\phi), \phi)$ is constant and equal to zero, we have that its derivative w.r.t. $\phi$ is also zero. Taking the total derivative we obtain

$$0 = \nabla_W^2 \mathcal{L}_T(W, \phi) \nabla_\phi W^*(\phi) + \nabla_\phi \nabla_W \mathcal{L}_T(W, \phi). \tag{6}$$

Multiplying by $\nabla_W^2 \mathcal{L}_T(W, \phi)^{-1}$ and reordering we obtain

$$\nabla_\phi W^*(\phi) = -\nabla_W^2 \mathcal{L}_T(W, \phi)^{-1} \nabla_\phi \nabla_W \mathcal{L}_T(W, \phi). \tag{7}$$

We can use this result to compute the gradients of the auxiliary set loss w.r.t $\phi$

$$\nabla_\phi \mathcal{L}_A(W^*(\phi)) = \nabla_W \mathcal{L}_A \cdot \nabla_\phi W^*(\phi) = -\nabla_W \mathcal{L}_A \cdot (\nabla_W^2 \mathcal{L}_T)^{-1} \cdot \nabla_\phi \nabla_W \mathcal{L}_T. \tag{8}$$

As discussed in the main text, fully optimizing $W$ to convergence is too computationally expensive. Instead, we update $\phi$ once for every several update steps for $W$, as seen in Alg. 1. To compute the vector inverse-Hessian product, we use Alg. 2 that was proposed in (Lorraine et al., 2020).

## B  EXPERIMENTAL DETAILS

### B.1  CUB 200-2011

**Data.** To examine the effect of varying training set sizes we use all 5994 predefined images for training according to the official split and, we split the predefined test set to 2897 samples for validation and 2897 for testing. All images were resized to $256 \times 256$ and Z-score normalized. During training, images were randomly cropped to 224 and flipped horizontally. Test images were centered cropped to 224. The same processing was applied in all fine-grain experiments.

**Training details for baselines.** We fine-tuned a ResNet-18 (He et al., 2016) pre-trained on ImageNet (Deng et al., 2009) with a classification layer on top for all tasks. Because the scale of auxiliary losses differed from that of the main task, we multiplied each auxiliary loss, on all compared method, by the scaling factor $\tau = 0.1$. It was chosen based on a grid search over $\{0.1, 0.3, 0.6, 1.0\}$ using the *Equal* baseline. We applied grid search over the learning rates in $\{1e-3, 1e-4, 1e-5\}$ and the weight decay in $\{5e-3, 5e-4, 5e-5\}$. For DWA (Liu et al., 2019b), we searched over the temperature in $\{0.5, 2, 5\}$ and for GradNorm (Chen et al., 2018), over $\alpha$ in $\{0.3, 0.8, 1.5\}$. The computational complexity of GSC (Du et al., 2018) grows with the number of tasks. As a result, we were able to run this baseline only in a setup where there are two loss terms: the main and the sum of all auxiliary tasks. We ran each configuration with 3 different seeds for 100 epochs with ADAM optimizer (Kingma and Ba, 2014) and used early stopping based on the validation set.

**The auxiliary set and auxiliary network.** In our experiments, we found that allocating as little as 20 samples from the training set for the auxiliary set and using a NN with 5 layers and 10 units in each layer yielded good performance for both deep linear and non-linear models. We found that our method was not sensitive to these design choices. We use skip connection between the main loss $\ell_{main}$ and the overall loss term and Softplus activation.

**Optimization of the auxiliary network.** In all variants of our method, the auxiliary network was optimized using SGD with 0.9 momentum. We applied grid search over the auxiliary network learning rate in $\{1e-2, 1e-3\}$ and weight decay in $\{1e-5, 5e-5\}$. The total training time of all methods was 3 hours on a 16GB Nvidia V100 GPU.

### B.2  NYUv2

The data consists of 1449 RGB-D images, split into 795 train images and 654 test images. We further split the train set to allocate 79 images, 10% of training examples, to construct a validation

set. Following (Liu et al., 2019b), we resize images to $288 \times 384$ pixels for training and evaluation and use SegNet (Badrinarayanan et al., 2017) based architecture as the backbone.

Similar to (Liu et al., 2019b), we train the model for 200 epochs using Adam optimizer (Kingma and Ba, 2014) with learning rate $1e-4$, and halve the learning rate after 100 epochs. We choose the best model with early stopping on a pre-allocated validation set. For DWA (Liu et al., 2019b) we set the temperature hyperparameter to 2, as in the NYUv2 experiment in (Liu et al., 2019b). For GradNorm (Chen et al., 2018) we set $\alpha = 1.5$. This value for $\alpha$ was used in (Chen et al., 2018) for the NYUv2 experiments. In all variants of our method, the auxiliary networks are optimized using SGD with 0.9 momentum. We allocate $2.5\%$ of training examples to form an auxiliary set. We use grid search to tune the learning rate $\{1e-3, 5e-4, 1e-4\}$ and weight decay $\{1e-5, 1e-4\}$ of the auxiliary networks. Here as well, we use skip connection between the main loss $\ell_{main}$ and the overall loss term and Softplus activation.

### B.3 LEARNING AUXILIARIES

**Multi-class classification datasets.** On the CIFAR datasets, we train the model for 200 epochs using SGD with momentum 0.9, weight decay $5e-4$, and initial learning rates $1e-1$ and $1e-2$ for CIFAR10 and CIFAR100, respectively. For the SVHN experiment, we train for 50 epochs using SGD with momentum 0.9, weight decay $5e-4$, and initial learning rates $1e-1$. The learning rate is modified using a cosine annealing scheduler. We use VGG-16 (Simonyan and Zisserman, 2014) based architecture for the CIFAR experiments, and a 4-layer ConvNet for the SVHN experiment. For MAXL (Liu et al., 2019a) label generating network, we tune the following hyperparameters: learning rate $\{1e-3, 5e-4\}$, weight decay $\{5e-4, 1e-4, 5e-5\}$, and entropy term weight $\{.2, .4, .6\}$ (see (Liu et al., 2019a) for details). We explore the same learning rate and weight decay for the auxiliary network in our method, and also tune the number of optimization steps between every auxiliary parameter update $\{5, 15, 25\}$, and the size of the auxiliary set $\{1.5\%, 2.5\%\}$ (of training examples). We choose the best model on the validation set and allow for early stopping.

**Fine-grain classification datasets.** In CUB experiments we use the same data and splits as described in Sections 5.2 and B.1. Oxford-IIIT Pet contains 7349 images of 37 species of dogs and cats. We use the official train-test split. We pre-allocate $30\%$ from the training set to validation. As a results, the total number of train/validation/test images are 2576/1104/3669 respectively. Cars (Krause et al., 2013) contains $16,185$ images of 196 car classes. We use the official train-test split and pre-allocate $30\%$ from the training set to validation. As a results, the total number of train/validation/test images are 5700/2444/8041 respectively. In all experiments we use ResNet-18 as the backbone network for both the primary and auxiliary networks. Importantly, the networks are not pre-trained. The task specific (classification) heads in both the primary and auxiliary networks is implemented using a 2-layer NN with sizes 512 and $C$. Where $C$ is number of labels (e.g., 200 for CUB and 37 for Oxford-IIIT Pet). In all experiments we use the same learning rate of $1e-4$ and weight decay of $5e-3$ which were shown to work best, based on a grid search applied on the STL baseline. For MAXL and AuxiLearn we applied a grid search over the auxiliary network learning rate and weight decay as described in the Multi-class classification datasets subsection. We tune the number of optimization steps between every auxiliary parameter update in $\{30, 60\}$ for Oxford-IIIT Pet and $\{40, 80\}$ for CUB and Cars. Also, the auxiliary set size was tuned over $\{0.084\%, 1.68\%, 3.33\%\}$ with stratified sampling. For our method, we leverage the module of AuxiLearn for combining auxiliaries. We use a Nonlinear network with either two or three hidden layers of sizes 10 (which was selected according to a grid search). The batch size was set to 64 in CUB and Cars experiments and to 16 in Oxford-IIIT Pet experiments. We ran each configuration with 3 different seeds for 150 epochs with ADAM optimizer and used early stopping based on the validation set.

## C  ADDITIONAL EXPERIMENTS

### C.1  IMPORTANCE OF AUXILIARY SET

In this section we illustrate the importance of the auxiliary set to complement our theoretical observation in Section 4. We repeat the experiment in Section 5.1, but this time we optimize the auxiliary parameters $\phi$ using the training data. Figure 5 shows how the tasks' weights change during training. The optimization procedure is reduced to single-task learning, which badly hurts

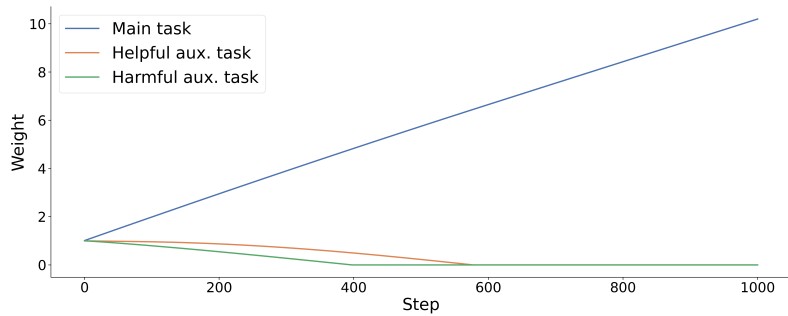

Figure 5: Optimizing task weights on the training set reduce to single-task learning.

generalization (see Figure 2). These results are consistent with (Liu et al., 2019a) that added an entropy loss term to avoid the diminishing auxiliary task.

### C.2 MONOTONOCITY

As discussed in the main text, it is a common practice to combine auxiliary losses as a convex combination. This is equivalent to parametrize the function $g(\boldsymbol{\ell}; \phi)$ as a linear combination over losses $g(\boldsymbol{\ell}; \phi) = \sum_{j=1}^{K} \phi_j \ell_j$, with non-negative weights, $\phi_j \geq 0$. Under this parameterization, $g$ is a monotonic non-decreasing function of the losses, since $\partial \mathcal{L}_T / \partial \ell_j \geq 0$. The non-decreasing property means that the overall loss grows (or is left unchanged) with any increase to the auxiliary losses. As a result, an optimization procedure that operates to minimize the combined loss also operates in the direction of reducing individual losses (or not changing them).

A natural question that arises is whether the function $g$ should generalize this behavior, and be constrained to be non-decreasing w.r.t. the losses as well? Non-decreasing networks can "ignore" an auxiliary task by zeroing its corresponding loss, but cannot reverse the gradient of a task by negating its weight. While monotonicity is a very natural requirement, in some cases, negative task weights (i.e., non-monotonicity) seem desirable if one wishes to "delete" input information not directly related to the task at hand (Alemi et al., 2017; Ganin and Lempitsky, 2015). For example, in domain adaptation, one might want to remove information that allows a discriminator to recognize the domain of a given sample (Ganin and Lempitsky, 2015). Empirically, we found that training with monotonic non-decreasing networks to be more stable and has better or equivalent performance, see Table 4 for comparison.

Table 4 compares monotonic and non-monotonic auxiliary networks in both the semi-supervised and the fully-supervised setting. Monotonic networks show a small but consistent improvement over non-monotonic ones. It is also worth mentioning that the non-monotonic networks were harder to stabilize.

Table 4: CUB 200-2011: Monotonic vs non-monotonic test classification accuracy ($\pm$ SEM) over three runs.

|  |  | Top 1 | Top 3 |
|---|---|---|---|
| 5-shot | Non-Monotonic | $46.3 \pm 0.32$ | $67.46 \pm 0.55$ |
|  | Monotonic | $\mathbf{47.07 \pm 0.10}$ | $\mathbf{68.25 \pm 0.32}$ |
| 10-shot | Non-Monotonic | $58.84 \pm 0.04$ | $77.67 \pm 0.08$ |
|  | Monotonic | $\mathbf{59.04 \pm 0.22}$ | $\mathbf{78.08 \pm 0.24}$ |
| Full Dataset | Non-Monotonic | $74.74 \pm 0.30$ | $88.3 \pm 0.23$ |
|  | Monotonic | $\mathbf{74.92 \pm 0.21}$ | $\mathbf{88.55 \pm 0.17}$ |

### C.3    NOISY AUXILIARIES

We demonstrate the effectiveness of AuxiLearn in identifying helpful auxiliaries and ignoring harmful ones. Consider a regression problem with main task $y = \mathbf{w}^T \mathbf{x} + \epsilon$, where $\epsilon \sim \mathcal{N}(0, \sigma^2)$. We learn this task jointly with $K = 100$ auxiliaries of the form $y_j = \mathbf{w}^T \mathbf{x} + |\epsilon_j|$, where $\epsilon_j \sim \mathcal{N}(0, j \cdot \sigma_{aux}^2)$ for $j = 1, ..., 100$. We use the absolute value on the noise so that noisy estimations are no longer unbiased, making the noisy labels even less helpful as the noise increases. We use a linear auxiliary network to weigh the loss terms. Figure 6 shows the learned weight for each task. We can see that the auxiliary network captures the noise patterns, and assign weights based on the noise level.

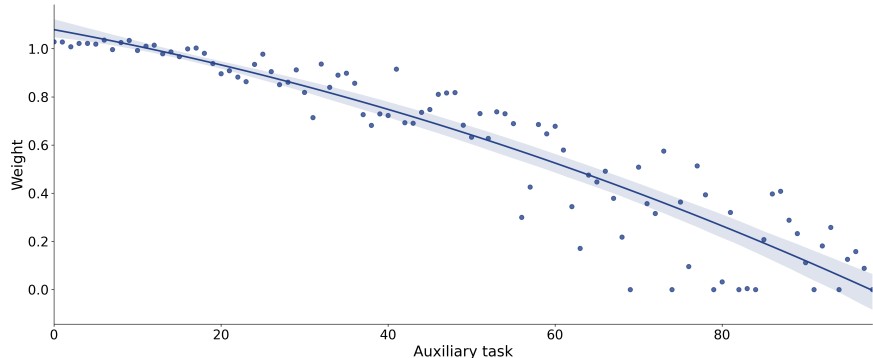

Figure 6: Learning with noisy labels: task ID is proportional to the label noise.

### C.4    CUB SENSITIVITY ANALYSIS

In this section, we provide further analysis for the experiments conducted on the CUB 200-2011 dataset in the 5-shot setup. We examine the sensitivity of a non-linear auxiliary network to the **size of the auxiliary set**, and the **depth of the auxiliary network**. In Figure 7a we test the effect of allocating (labeled) samples from the training set to the auxiliary set. As seen, allocating between $10 - 50$ samples results in similar performance picking at 20. The figure shows that removing too many samples from the training set can be damaging. Nevertheless, we notice that even when allocating 200 labeled samples (out of 1000), our nonlinear method is still better than the best competitor GSC (Du et al., 2018) (which reached an accuracy of $42.57$).

Figure 7b shows how accuracy changes with the number of hidden layers. As expected, there is a positive trend. As we increase the number of layers, the network expressivity increases, and the performance improves. Clearly, making the auxiliary network too large may cause the network to overfit the auxiliary set as was shown in Section 4, and empirically in (Lorraine et al., 2020).

### C.5    LINEARLY WEIGHTED NON-LINEAR TERMS

To further motivate the use of non-linear interactions between tasks, we train a linear auxiliary network over a polynomial kernel on the tasks segmentation, depth estimation and normal prediction from the NYUv2 dataset. Figure 8 shows the learned loss weights. From the figure, we learn that two of the three largest weights at the end of training belong to non-linear terms, specifically, $Seg^2$ and $Seg \cdot Depth$. Also, we observe a *scheduling* effect, in which at the start of training, the auxiliary network focuses on the auxiliary tasks (first $\sim 50$ steps), and afterwards it draws most of the attention of the primary network towards the main task.

### C.6    FIXED AUXILIARY

As a result of alternating between optimizing the primary network parameters and the auxiliary parameters, the weighting of the loss terms are updated during the training process. This means that the loss landscape is changed during training. This effect is observed in the illustrative examples

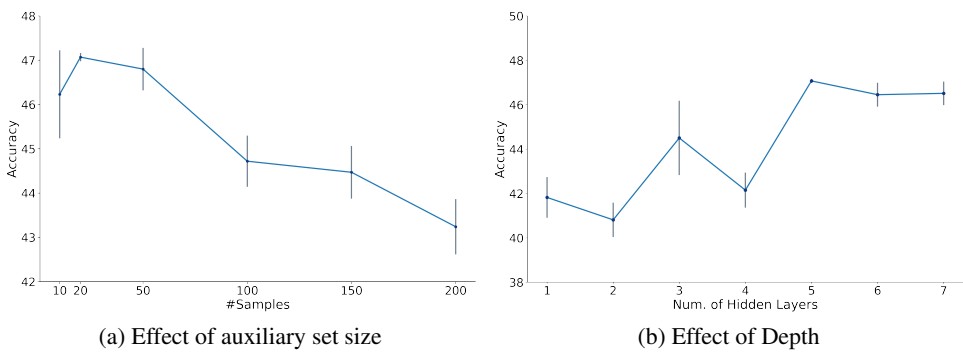

(a) Effect of auxiliary set size  (b) Effect of Depth

Figure 7: Mean test accuracy ($\pm$ SEM) averaged over 3 runs as a function of the number of samples in the auxiliary set (left) and the number of hidden layers (right). Results are on 5-shot CUB 200-2011 dataset.

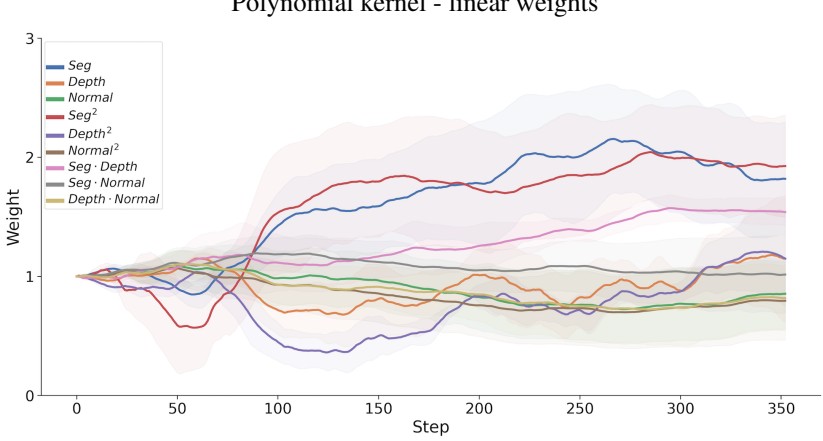

Figure 8: Learned linear weights for a polynomial kernel on the loss terms of the tasks segmentation, depth estimation and normal prediction from the NYUv2 dataset.

described in Section 5.1 and Section C.5, where the auxiliary network focuses on different tasks during different learning stages. Since the optimization is non-convex, the end result may depend not only on the final parameters but also on the loss landscape during the entire process.

We examined this effect with the following setup on the 5-shot setting on CUB 200-2011 dataset: we trained a non-linear auxiliary network and saved the best model. Then we retrain with the same configuration, only this time, the auxiliary network is initialized using the best model, and is kept fixed. We repeat this using ten different random seeds, affecting the primary network initialization and data shuffling. As a result, we observed a drop of 6.7% on average in the model performance with an std of 1.2% (46.7% compared to 40%).

## C.7 FULL CUB DATASET

In Section 5.2 we evaluated AuxiLearn and the baseline models performance under a semi-supervised scenario in which we have 5 or 10 labeled samples per class. For completeness sake, we show in Table 5 the test accuracy results in the standard fully-supervised scenario. As can be seen, in this case the STL baseline achieves the highest top-1 test accuracy while our nonlinear method is second on the top-1 and first on the top-3. Most baselines suffer from severe negative transfer due to the large number of auxiliary tasks (which are not needed in this case) while our method cause minimal performance degradation.

Table 5: CUB 200-2011: Fully supervised test classification accuracy ($\pm$ SEM) averaged over three runs.

|  | Top 1 | Top 3 |
|---|---|---|
| STL | **75.2 $\pm$ 0.52** | 88.4 $\pm$ 0.36 |
| Equal | 70.16 $\pm$ 0.10 | 86.87 $\pm$ 0.22 |
| Uncertainty | 74.70 $\pm$ 0.56 | 88.21 $\pm$ 0.14 |
| DWA | 69.88 $\pm$ 0.10 | 86.62 $\pm$ 0.20 |
| GradNorm | 70.04 $\pm$ 0.21 | 86.63 $\pm$ 0.13 |
| GSC | 71.30 $\pm$ 0.01 | 86.91 $\pm$ 0.28 |
| **AuxiLearn (ours)** | | |
| Linear | 70.97 $\pm$ 0.31 | 86.92 $\pm$ 0.08 |
| Deep Linear | 73.6 $\pm$ 0.72 | 88.37 $\pm$ 0.21 |
| Nonlinear | 74.92 $\pm$ 0.21 | **88.55 $\pm$ 0.17** |

## C.8 CITYSCAPES

Cityscapes (Cordts et al., 2016) is a high-quality urban-scene dataset. We use the data provided in (Liu et al., 2019b) with 2975 training and 500 test images. The data comprises of four learning tasks: 19-classes, 7-classes and 2-classes semantic segmentation, and depth estimation. We use the 19-classes semantic segmentation as the main task, and all other tasks as auxiliaries. We allocate $10\%$ of the training data for validation set, to allow for hyperparameter tuning and early stopping. We further allocate $2.5\%$ of the remaining training examples to construct the auxiliary set. All images are resized to $128 \times 256$ to speed up computation.

We train a SegNet (Badrinarayanan et al., 2017) based model for $150$ epochs using Adam optimizer (Kingma and Ba, 2014) with learning rate $1e-4$, and halve the learning rate after $100$ epochs. We search over weight decay in $\{1e-4, 1e-5\}$. We compare AuxiLearn to the same baselines used in Section 5.2 and search over the same hyperparameters as in the NYUv2 experiment. We set the DWA temperature to 2 similar to (Liu et al., 2019b), and the GradNorm hyperparameter $\alpha$ to 1.5, as used in (Chen et al., 2018) for the NYUv2 experiments. We present the results in Table 6. The ConvNet variant of the auxiliary network achieves best performance in terms of mIoU and pixel accuracy.

Table 6: 19-classes semantic segmentation test set results on Cityscapes, averaged over three runs ($\pm$ SEM).

|  | mIoU | Pixel acc. |
|---|---|---|
| STL | 30.18 $\pm$ 0.04 | 87.08 $\pm$ 0.18 |
| Equal | 30.45 $\pm$ 0.14 | 87.14 $\pm$ 0.08 |
| Uncertainty | 30.49 $\pm$ 0.21 | 86.89 $\pm$ 0.07 |
| DWA | 30.79 $\pm$ 0.32 | 86.97 $\pm$ 0.26 |
| GradNorm | 30.62 $\pm$ 0.03 | 87.15 $\pm$ 0.04 |
| GCS | 30.32 $\pm$ 0.23 | 87.02 $\pm$ 0.12 |
| **AuxiLearn (ours)** | | |
| Linear | 30.63 $\pm$ 0.19 | 86.88 $\pm$ 0.03 |
| Nonlinear | 30.85 $\pm$ 0.19 | 87.19 $\pm$ 0.20 |
| ConvNet | **30.99 $\pm$ 0.05** | **87.21 $\pm$ 0.11** |

## C.9 LEARNING SEGMENTATION AUXILIARY FOR 3D POINT CLOUDS

Recently, several methods were offered for learning auxiliary tasks in point clouds (Achituve et al., 2020; Hassani and Haley, 2019; Sauder and Sievers, 2019); however, this domain is still largely unexplored and it is not yet clear which auxiliary tasks could be beneficial beforehand. Therefore, it is desirable to automate this process, even at the cost of performance degradation to some extent compared to human designed methods.

We further evaluate our method in the task of generating helpful auxiliary tasks for 3D point-cloud data. We propose to extend the use of AuxiLearn for segmentation tasks. In Section 5.4 we trained an auxiliary network to output soft auxiliary labels for classification task. Here, we use a similar

Table 7: Learning auxiliary segmentation task. Test mean IOU on ShapeNet part dataset averaged over three runs (±SEM) - 30 shot

|  | Mean | Airplane | Bag | Cap | Car | Chair | Earphone | Guitar | Knife | Lamp | Laptop | Motorbike | Mug | Pistol | Rocket | Skateboard | Table |
|---|---|---|---|---|---|---|---|---|---|---|---|---|---|---|---|---|---|
| Num. samples | 2874 | 341 | 14 | 11 | 158 | 704 | 14 | 159 | 80 | 286 | 83 | 51 | 38 | 44 | 12 | 31 | 848 |
| STL | 75.6 | 68.7 | **82.9** | 85.2 | **65.6** | 82.3 | 70.2 | 86.1 | 75.1 | 68.4 | 94.3 | 55.1 | 91.0 | 72.6 | 60.2 | 72.3 | 74.2 |
| DAE | 74.0 | 66.6 | 77.6 | 79.1 | 60.5 | 81.2 | **73.8** | 87.1 | 77.0 | 65.4 | 93.6 | 51.8 | 88.4 | **74.0** | 55.4 | 68.4 | 72.7 |
| DefRec | 74.6 | 68.6 | 81.2 | 83.8 | 63.6 | 82.1 | 72.9 | 86.9 | 72.7 | 69.4 | 93.4 | 51.8 | 89.7 | 72.0 | 57.2 | 70.5 | 71.7 |
| RS | **76.5** | **69.7** | 79.1 | **85.9** | 64.9 | **83.8** | 68.4 | 82.8 | 79.4 | **70.7** | 94.5 | **58.9** | 91.8 | 72.0 | 53.4 | 70.3 | **75.0** |
| AuxiLearn | 76.2 | 68.9 | 78.3 | 83.6 | 64.9 | 83.4 | 69.7 | **87.4** | **80.7** | 68.3 | **94.6** | 53.2 | **92.1** | 73.7 | **61.6** | **72.4** | 74.6 |

approach, assigning a soft label vector to each point. We then train the primary network on the main task and the auxiliary task of segmenting each point based on the learned labels.

We evaluated the above approach in a part-segmentation task using the ShapeNet part dataset (Yi et al., 2016). This dataset contains 16,881 3D shapes from 16 object categories (including Airplane, Bag, Lamp), annotated with a total of 50 parts (at most 6 parts per object). The main task is to predict a part label for each point. We follow the official train/val/test split scheme in (Chang et al., 2015). We also follow the standard experimental setup in the literature, which assumes known object category labels during segmentation of a shape (see e.g., (Qi et al., 2017; Wang et al., 2019)). During training we uniformly sample 1024 points from each shape and we ignore points normal. During evaluation we use all points of a shape. For all methods (ours and baselines) we used the DGCNN architecture (Wang et al., 2019) as the backbone feature extractor and for part segmentation. We evaluated performance using point-Intersection over Union (IoU) following (Qi et al., 2017).

We compared AuxiLearn with the following baselines: **(1) Single Task Learning (STL):** Training with the main task only. **(2) DefRec:** An auxiliary task of reconstructing a shape with a deformed region (Achituve et al., 2020). **(3) Reconstructing Spaces (RS):** An auxiliary task of reconstructing a shape from a shuffled version of it (Sauder and Sievers, 2019). and **(4) Denoising Auto-encoder (DAE):** An auxiliary task of reconstructing a point-cloud perturbed with an iid noise from $\mathcal{N}(0, 0.01)$.

We performed hyper-parameter search over the primary network learning rate in $\{1e - 3, 1e - 4\}$, weight decay in $\{5e - 5, 1e - 5\}$ and weight ratio between the main and auxiliary task of $\{1 : 1, 1 : 0.5, 1 : 0.25\}$. We trained each method for 150 epochs, used the Adam optimizer with cosine scheduler. We applied early stopping based on the mean IoU of the validation set. We ran each configuration with 3 different seeds and report the average mean IOU along with the SEM. We used the segmentation network proposed in (Wang et al., 2019) with an exception that the network wasn't supplied with the object label as input.

For AuxiLearn, we used a smaller version of PointNet (Qi et al., 2017) as the auxiliary network without input and feature transform layers. We selected PointNet because its model complexity is light and therefore is a good fit in our case. We learned a different auxiliary task per each object category (with 6 classes per category) since it showed better results. We performed hyper-parameter search over the auxiliary network learning rate in $\{1e - 2, 1e - 3\}$, weight decay in $\{5e - 3, 5e - 4\}$. Two training samples from each class were allocated for the auxiliary set.

Table 7 shows the mean IOU per category when training with only 30 segmented point-clouds per object category (total of 480). As can be seen, AuxiLearn performance is close to RS (Sauder and Sievers, 2019) and improve upon other baselines. This shows that in this case, our method generates useful auxiliary tasks that has shown similar or better gain than those designed by humans.

## C.10 LEARNING AN AUXILIARY CLASSIFIER

In Section 5.4 we show how AuxiLearn learns a novel auxiliary to improve upon baseline methods. For the fine-grained classification experiments, we use only 30 samples per class. Here we also compare AuxiLearn with the baseline methods when there are only 15 images per class. Table 8 shows that AuxiLearn is superior to baseline methods in this setup as well, even though it requires to allocate some samples from the training data to the auxiliary set.

To further examine the effect of learning novel auxiliary task with varying train set size, we provide here additional experiments on the CIFAR10 dataset. We evaluate the methods with of 10%, 15% and 100% of training examples. The results are presented in Table 9. As expected, learning with

Table 8: Learning auxiliary task. Test accuracy averaged over three runs (±SEM) - 15 shot

|  | CUB | Pet |
|---|---|---|
| STL | $22.6 \pm 0.2$ | $13.6 \pm 0.7$ |
| MAXL-F | $24.2 \pm 0.7$ | $14.1 \pm 0.1$ |
| MAXL | $24.2 \pm 0.8$ | $14.2 \pm 0.2$ |
| **AuxiLearn** | $\mathbf{26.1 \pm 0.7}$ | $\mathbf{18.0 \pm 0.9}$ |

Table 9: CIFAR10 test results accuracy averaged over three runs (±SEM).

|  | CIFAR10 | | |
|---|---|---|---|
|  | 10% | 15% | 100% |
| STL | $72.63 \pm 2.14$ | $80.30 \pm 0.09$ | $93.36 \pm 0.05$ |
| MAXL | $75.85 \pm 0.32$ | $81.37 \pm 0.26$ | $93.49 \pm 0.02$ |
| **AuxiLearn** | $\mathbf{76.75 \pm 0.08}$ | $\mathbf{81.42 \pm 0.30}$ | $\mathbf{93.54 \pm 0.05}$ |

auxiliaries is mostly helpful in the low data regime. Nonetheless, AuxiLearn improves over single task learning and MAXL for all training set sizes.

## D THEORETICAL CONSIDERATIONS

In this section, we discuss the theoretical limitations of AuxiLearn. First, we discuss the smoothness of our loss criterion while learning to combine losses using DNNs. Next, we present limitations that may arise from utilizing the IFT and their resolution. Finally, we discuss the approximations made for achieving an efficient optimization procedure.

**Smoothness of the loss criterion.** When learning to combine losses as described in Section 3.2, one must take into consideration the smoothness of the learn loss criterion as a function of $W$. This limits, at least in theory, the design choice of the auxiliary network. In our experiments we use smooth activation functions, namely Softplus, to ensure the existence of $\partial \mathcal{L}_T / \partial W$. Nonetheless, using non-smooth activation (e.g. ReLU) results with a piecewise smooth loss function hence might work well in practice.

**Assumptions for IFT.** One assumption for applying the IFT as described in Section 3.4, is that $\mathcal{L}_T$ is continuously differentiable w.r.t to the auxiliary and primary parameters. This assumption limits the design choice of both the auxiliary, and the primary networks. For instance, one must utilize only smooth activation functions. However, many non-smooth components can be replaced with smooth counterparts. For example, ReLU can be replaced with Softplus, $ReLU(x) = \lim_{\alpha \to \infty} \ln(1 + \exp(\alpha x))/\alpha$, and the beneficial effects of Batch-Normalization can be captured with Weight-Normalization as argued in (Salimans and Kingma, 2016).

For the setup of *learning to combine losses*, we use the above substitutes, namely Softplus and Weight Normalization, however for the *learning a novel auxiliary* setup, we share architecture between primary and auxiliary network (e.g. ResNet18). While using non-smooth components may, in theory, cause issues, we show empirically through extensive experiment that AuxiLean performs well in practice, and its optimization is stable. Furthermore, we note that while ReLUs are non-smooth, they are piecewise smooth, hence the set of non-smoothness points is a zero-measure set.

**Approximations.** Our optimization procedure relies on several approximations to efficiently solve complex bi-level optimization. This trade-off between computation efficiency and accurate approximation can be controlled by (i) The number of Neumann series components, and; (ii) The number of optimization steps between auxiliary parameters update. While we cannot guarantee that the bi-level optimization process converges, empirically we observe a stable optimization process.

Our work builds on previous studies in the field of hyperparameter optimization (Lorraine et al., 2020; Pedregosa, 2016). Lorraine et al. (2020) provide an error analysis for both approximations, in a setup for which the exact Hessian can be evaluated in closed form. We refer the readers to Pedregosa (2016) for theoretical analysis and results regarding the second approximation (i.e. sub-optimally of the inner optimization problem in Eq. 2).

