# OpenReview forum: "Auxiliary Learning by Implicit Differentiation"
_ICLR.cc/2021/Conference — ICLR 2021 Poster_

### Official Review · AnonReviewer4 · 2020-10-27
**Interesting Approach to Multi-task learning using auxiliary tasks with strong empirical results**

**Rating:** 7
**Confidence:** 3

**Review:**

The paper proposes AuxiLearn, a framework that can be used to combine losses from multiple auxiliary tasks (if present) into a single combined, loss function that does not require expensive grid search over possible linear combination. It uses an implicit differentiation-based approach to train a (deep) non-linear network that weighs the various auxiliary losses to optimize the generalization capabilities of the network. In the absence of such pre-defined tasks, a variation of the approach, using teacher-student networks, helps create relevant tasks to improve the performance of the network. Experiments across tasks such as classification (both few-shot and with limited labels) and segmentation show that the approach helps improve the performance of the model on the main task.

Comments:
- There are some issues with notations and the overall presentation that make it a little hard to follow. For example, in section 5.1, why are there different weights/parameters (w* and \tilde{w}) for the helpful and harmful tasks? Should the variation not just be w.r.t epsilon? Similarly, in Section 4.1, while the analysis is nice, how is this used in the proposed framework? Is the Newton update monitored in the proposed framework to identify the more important auxiliary task? If so, then I think that should be the core contribution since there can exist a non-trivial number of auxiliary tasks (e.g. 312 in the CUB dataset), and identifying them would greatly reduce the complexity of training. If the Newton update is not used to "prune" tasks, I am not convinced that it adds value to the overall theme of the paper and the space could be used to describe the evaluation of the approach in more detail (as done in the supplementary).
- Ablation studies are missing that could add more value to the evaluation and help highlight the key contributions. For example, what is the effect of depth on the deep auxiliary network? It is mentioned that a 5 layer network with 10 neurons each was used. Ablations on this configuration would help visualize the effect of network complexity on the aux loss weights.
- One question that arises is whether the aux losses are scaled based on the amount of data available for each of these tasks. For example, in CUB-200 2011, although there are 312 tasks, only around 130 "tasks" or attributes that have a significant amount of annotations for at least 5 of the classes. Does this have any effect on the combined loss?

---

> ### Author Response · Authors · 2020-11-16
> **Response to Reviewer 4**
>
> Thank you for the helpful feedback and insightful comments.
>
> **In Section 5.1 why are there different parameters for the helpful and harmful tasks?**
> The weights of the helpful task should help to guide the learning of the main task. Therefore, they are the same as the ones of the main task ($w^*$) but with a different noise level. In this experiment, the noise level of the main task is higher than the noise level of the auxiliary tasks. Parameters of the harmful task ($\tilde{w}$) indeed are different from those of the main and helpful auxiliary. This setup emphasizes how AuxiLearn learns to rely on the useful auxiliary task and ignore the harmful one.
>
> **How is the analysis in Section 4.2 used in the proposed framework? Can the Newton update be used to prune tasks?**
> The theoretical analysis reveals a connection between the AuxiLearn update step, and the Newton update, in a specific case where the auxiliary network is linear, and the auxiliary weight equals zero. In this case, an alignment between the directions of the gradients of the main task and the Newton update suggests the auxiliary is helpful. Hence, we do not directly use the Newton update. Generally, it is hard to assess the importance of an auxiliary task because it may change throughout the optimization process, as can be seen in Appendix C.5. Nonetheless, we examined the learned tasks’ weights in our analysis as described below.
>
> For a linear auxiliary network, one can view the final learned weights as an indicator of the importance of the tasks. In Appendix C.3 we show that AuxiLearn can learn meaningful weights in that sense. The learned weights are proportional to the label noise of the tasks. Tasks with extreme label noise are “pruned” with zero weight.
>
> The non-linear case is harder to analyze. An attempt to demonstrate this case is depicted in Appendix C.5 in which we show a linear combination of the losses over a polynomial kernel (thus forming a non-linear combination of the original losses). In the general case, a non-linear combination of losses can be viewed as an adaptive linear weighting, where losses have a different set of weights for each datum that is determined by the gradient of $L_T$ w.r.t the task loss. We added a visualization of the learned per-pixel weights that demonstrates that in section 5.3 (which was in Appendix C.2 in the original submission).
>
> **Ablation studies are missing.**
> .Ablation studies are provided in the supplementary. Specifically, Appendix C.4 shows the effect of the auxiliary set size and the auxiliary network depth for the case of combining losses on the CUB dataset. Also, Appendix C.2 compares monotonic vs. non-monotonic auxiliary networks.
>
> **Does the auxiliary losses scale based on the amount of data available for the tasks?**
> AuxiLearn learns to weigh tasks based on their impact on the generalization abilities of the primary network. Depending on the data, common or rare tasks could help the model to be more discriminative on the main task. Following this comment, we computed the Pearson and Spearman correlation coefficients between a linear model tasks’ weights and the frequency of the auxiliary classes. Both were ~0.35 on average across 3 seeds. This shows an intermediate correlation between the weights and the frequency of the tasks.

---

> > ### Comment · AnonReviewer4 · 2020-11-24
> > **Thanks for the clarification**
> >
> > Thanks for providing the clarifications. I really appreciate your detailed responses. Having read the other reviews and the author's responses, I feel that the paper makes a good contribution. Overall, I think this is a good paper and would recommend its acceptance. I stand by my original rating.

---

### Official Review · AnonReviewer3 · 2020-10-29
**This paper proposes to make use of implicit differerntiation to improve auxiliary learning.**

**Rating:** 6
**Confidence:** 3

**Review:**

This paper proposes to make use of implicit differerntiation to improve auxiliary learning, including learning to combine the manually designed auxiliary tasks and learning new auxiliary tasks.

Reasons for scores: Overall the paper looks interesting and technically sound. However, strong experiments could have been done for being more convincing.

Overall the paper is clearly written and technically sound.  The problem studied is significant and should be of interest to a broad range of people in the machine learning community. Some concerns are listed below.

First, I have concerns on the learning new auxiliary tasks, which is formulated as student-teacher maner. This step sounds somewhat counter-intuitive. The initial teacher network might simply produce meaningless labels which are used to guide the student network; it begs the question that what is the intuitition behind that pushs both the teacher and student network to learn non-trivial solutions.

In general the experiments could have been stronger. Although the authors compare with other multi-task works such as using uncertainty as  a way to combine loss, comparisons with the current best results in each benchmark task is desired as well. This will help reveal where the proposed method stands in terms of accuracy.

In sec 5.4, it demonstrates the utility of learnable auxiliary in the low-data regime and only 5% of the training data are used. This seems to indicate that the method is only useful in the case of scarce training data. The authors could have reported results under different amount of training data; this is important for readers to understand the range within which the method is expected to help.

---

> ### Author Response · Authors · 2020-11-16
> **Response to Reviewer 3**
>
> Thank you for the helpful feedback and insightful comments.
>
> **Motivation and intuition for learning a novel auxiliary task.**
> We first copy here our response to R2 which has a related comment, and then we address the more specific point raised in this comment.
>
> To explain the intuition behind learned auxiliaries, we first note that auxiliary tasks play the role of smart regularizers. They induce a bias over the representation that is shared with the main task, pushing it to generalize better.  When an auxiliary task is given, it reflects known priors about aspects of the representation that would help generalization. When it is not given, we argue that it is feasible to learn which aux tasks help generalize using an (auxiliary) validation set. We show empirically that this improved generalization carries well to a test set. To put it simply, our approach optimizes auxiliary tasks by how well they help generalization to the auxiliary validation set. This learned regularization makes the joint representation generalize better.
>
> Indeed, AuxiLearn significantly improves upon the single-task learning, and upon MAXL-F, an approach that generates random labels that are indeed meaningless. These results suggest that AuxiLearn learns meaningful auxiliary tasks that help generalize.
>
> We illustrated that effect with a visualization of the soft labels learned by the auxiliary network. This was given in Appendix C.10 of the submitted version and now moved to Figure 4 in the main text. We used tSNE to project the labels generated by the auxiliary network within the same class of the main task to 2D. The projection reveals clear patterns with a finer partition among same-class images. For example, we show that the labels learned for the Deer class capture complex, semantic features. The middle box in the figure under the Deer class contains deers with antlers in various poses and varying backgrounds.
>
> More specifically, regarding why the teacher does not specify random labels. AuxiLearn finds those labels that make the main task generalize better to the auxiliary validation set. Specifically, there is an incentive for the auxiliary network (parametrized by $\phi$)  to generate non-trivial labels. The auxiliary network is optimized to increase the generalization performance of the primary network, measured by the main task loss over the auxiliary set. The auxiliary network can steer the learned representation only through the labels that it generates. Thus, to force the primary network to generalize better to the auxiliary set, the auxiliary network must create a meaningful task.
>
> Following this comment, we made the following changes to the paper: (1) We motivate better the role of learned auxiliaries as regularizers in section 3.3; (2) The visualization of learned labels is now given in Figure 4 of the main text, moved from Appendix C.10.
>
> **A comparison to SoTA in each benchmark task is desired.**
> Most experiments in this paper focus on the limited-data regime. In these cases, there is no clear SOTA to compare with. Approaches that achieve SoTA on benchmarks often do that by integrating many small improvements, from data augmentation to extensive parameter tuning and lengthy training. The experiments in this paper follow the benchmark in that specific community and are designed to isolate the effect of learning with auxiliaries.
>
> **Applicability of learning an auxiliary outside the low-data regime.**
> Auxiliary tasks act as smart regularizers, so they are expected to be most beneficial when learning with limited data where regularization is critical. Following this helpful suggestion, we examine the utility of learned auxiliary with varying sizes of training data. We add new experiments in Table 9, appendix C.10, to evaluate AuxiLearn training with 10%, 15%, and 100% of CIFAR10 training samples. AuxiLearn improves the test accuracy over baselines for all training set sizes, but the effect is largest in the low-data regime. Also, note that our experiments on CUB and Cars used all training samples.

---

### Official Review · AnonReviewer5 · 2020-11-09
**Overall, It is an okay submission with rich experimental results; however, some main part problems degrade the score.**

**Rating:** 6
**Confidence:** 3

**Review:**

This paper proposes a way to combine the auxiliary tasks' losses. Moreover, when the auxiliary task is unknown, a DNN is utilized to learn the auxiliary task automatically.

1. For the auxiliary tasks combing cases, how to ensure the g is smooth w.r.t. the main task parameters W, i.e., does pdiv{L_T}/pdiv{W} exists? When g is DNN, some non-smooth activation function will break down the framework and make the main task parameters hard to learn.
2. As for the IFT, one assumption or say the requirement for Eq. (3) to hold is the function div{L_T}/div{W} should be continuously differentiable w.r.t. {W, \phi}. This will limit the design of the DNN f and g, such as we cannot use BN and non-smooth activation function (e.g., ReLU, etc.). Although the code and the automatic differentiation tools may still work in practice, it at least wrong from the theoretical perspective.
3. Since several approximations steps are utilized (approximation W^* and inexact vector and Hessian inverse product), it is necessary and better to provide the error analysis since the convergence of the bi-level optimization is not obvious.
4. I do not see much insight from Sec. 4.2. If d{L_A}/d{\phi} > 0 means the auxiliary task is helpless, does it mean that I can make it helpful by assigning it a negative weight say \phi_i? But this is in contradiction with the experimental results in which the monotonic networks tend to have a better performance.

minor comments:
1. Some statements are inconsistent with (a) in figure 1. The subfigure (a) in Fig.1 seems to show that L_T is g(\cdot;\phi), while the end at page 3 states 'L_T = l_mian + g'. It is confused here, since subfigure (b) shows the way of 'L_T = l_mian + g'.

---

> ### Author Response · Authors · 2020-11-16
> **Response to Reviewer 5**
>
> Thank you for the helpful feedback and insightful comments.
>
> **Ensure the $g$ is smooth w.r.t. the main task parameters W.**
> Indeed, non-smooth activation may result in a non-smooth loss criterion. To address this, our experiments use Softplus activations. When combined with a monotone auxiliary network, Softplus creates a smooth loss criterion. The loss criterion is also smooth for the setup of learning auxiliaries.
>
> **IFT renders design limitations of $f$ and $g$. Therefore, BN and non-smooth activation functions cannot be used.**
> We agree that using BN and non-smooth activation functions would not reconcile with the theoretical assumptions. For the setting of “learning to combine auxiliaries”, we use smooth activations (Softplus) and weight normalization as a substitute for batch normalization. For the setting of “learning novel auxiliaries”, we consider a similar architecture for the primary and auxiliary networks, e.g. ResNet18. While using non-smooth activations may, in theory, cause issues, we show empirically through extensive experiment that AuxiLean performs well in practice, and its optimization is stable. Note that even though ReLUs are not smooth, they are piecewise smooth, so the set of non-smoothness points is a zero-measure set.
>
> In general, as stated above, many non-smooth components can be replaced with smooth counterparts. For example, ReLU can be replaced with Softplus ($\lim_{\alpha\to\infty}\ln{(1+\exp(\alpha x))/\alpha=ReLU(x)}$), and the beneficial effects of Batch Norm can be captured with Weight Normalization (as argued in Salimans & Kingma, 2016).
>
> **Convergence of the bi-level optimization is not obvious due to approximations made.**
> Our optimization procedure relies on several approximations to efficiently solve complex bi-level optimization. This trade-off between computation efficiency and accurate approximation can be controlled by (1) The number of Neumann series components; and (2) The number of optimization steps between auxiliary parameters update.
>
> While we cannot guarantee that the bi-level optimization process converges, empirically we observe a stable optimization process.
>
> Our work builds on previous work in the field of hyperparameter optimization (Lorraine et al., 2020). In that study, the authors provide an error analysis for both approximations, in a setup for which the exact Hessian can be evaluated in closed form.
>
>
> **Will it be possible to make an auxiliary task helpful by assigning to it a negative weight? If so, it is in contradiction to the empirical results on monotonic vs non-monotonic auxiliary networks.**
> Our theoretical analysis characterizes helpful auxiliaries under the convention that the weight for a helpful auxiliary should be positive. If the weight of a loss term is negative, the learner is encouraged to perform worse on that task and “erase all meaningful information” for this task. However, our theoretical results do not contradict our empirical findings: indeed, non-monotonic networks could learn monotonic mappings, and have the potential to outperform monotonic networks. In practice, we found the optimization process for monotonic networks to be much more stable, making the solutions obtained for monotonic nets perform better.
>
> **Fig. 1a and the text do not match:**
> Thank you. We revised the figure.
>
> **Citations:**
>
> Salimans, T., & Kingma, D. P. (2016). Weight normalization: A simple reparameterization to accelerate training of deep neural networks. In Advances in neural information processing systems (pp. 901-909).
>
> Lorraine, J., Vicol, P., & Duvenaud, D. (2020, June). Optimizing millions of hyperparameters by implicit differentiation. In International Conference on Artificial Intelligence and Statistics (pp. 1540-1552). PMLR.

---

### Official Review · AnonReviewer2 · 2020-11-11
**The intuition makes sense. Comprehensive experiments are done.**

**Rating:** 6
**Confidence:** 3

**Review:**

This paper studies a variant of multi-task learning, auxiliary learning, where one main task dominates, and other tasks are used to learn a good representation. To achieve this goal, the authors propose a learning-to-learn algorithm. In particular, the auxiliary losses are represented by a vector and then transformed to a new loss term via linear or nonlinear function $h$. They also made two more contributions. First, an approach of new auxiliary task generation is proposed. Second, an implicit differentiation based optimization method is proposed to find the solution. Both theoretical analysis and empirical studies demonstrate the superiority of their proposed model.

Pros:
1. The intuition makes sense. When the main task dominates, its generalization can help guide the weighting of the weights of auxiliary tasks.
2. The theoretical analysis further supports the claim of this work.
3. Comprehensive experiments are made to verify the effectiveness of the proposal.

Cons:
1. The idea of learning new auxiliary tasks is wired and less intuitive. The learning of the whole system is still the main task loss, without any useful supervision (or self-supervision). Therefore, it is doubtful whether the learned task is meaningful, or may involve some chaos in the system. Overall, I think this part does harm to this work and the authors may consider removing it.
2. Notations are sort of unclear. In section 3.2, $\mathbf{l}$ is a $K+1$-dim vector, but it seems $g(\mathbf{l};\phi)$ only maps the $K$ auxiliary tasks' losses to a scalar.
3. There are some typos and the authors need to polish this paper again.

---

> ### Author Response · Authors · 2020-11-16
> **Response to Reviewer 2**
>
> Thank you for the helpful feedback and insightful comments.
>
> **Motivation and intuition for learning a novel auxiliary task.**
> To explain the intuition behind learned auxiliaries, we first note that auxiliary tasks play the role of smart regularizers. They induce a bias over the representation that is shared with the main task, pushing it to generalize better.  When an auxiliary task is given, it reflects known priors about aspects of the representation that would help generalization. When it is not given, we argue that it is feasible to learn which aux tasks help generalize using an (auxiliary) validation set. We show empirically that this improved generalization carries well to a test set. To put it simply, our approach optimizes auxiliary tasks by how well they help generalization to the auxiliary validation set. This learned regularization makes the joint representation generalize better.
>
> Indeed, AuxiLearn significantly improves upon the single task learning and MAXL-F which generates random labels that are indeed meaningless. These results suggest that AuxiLearn learns a meaningful auxiliary task that helps generalize.
>
> We illustrated that effect with a visualization of the soft labels learned by the auxiliary network. This was given in Appendix C.10 of the submitted version and now moved to Figure 4 in the main text. We used tSNE to project the labels generated by the auxiliary network within the same class of the main task to 2D. The projection reveals clear patterns with a finer partition among same-class images. For example, we show that the labels learned for the Deer class capture complex, semantic features. The middle box in the figure under the Deer class contains deers with antlers in various poses and varying backgrounds.
>
> Following this comment and R3 comment, we: (1) Motivate better the role of learned auxiliaries as regularizers in section 3.3; (2) We moved the illustration in Appendix C.10 to the main body of the paper.
>
> **Unclear notation in Sec. 3.2.**
> Note that $g(\ell;\phi)$ maps a $K+1$ vector (main loss + K auxiliary losses) to a scalar. In this manner, $g$ can learn complex interactions between the main and auxiliary losses. The overall loss ($L_T$) is then the main loss + the output from $g$.
>
> **Minor comments and typos.**
> Thank you. We fixed all typos and grammar issues.

---

### Official Review · AnonReviewer6 · 2020-12-08
**Good Paper with Sound Technique and Sufficient  Experimental Support**

**Rating:** 7
**Confidence:** 3

**Review:**

This paper pinpoints the key issues of Auxiliary Learning: (1). how to design useful auxiliary tasks, (2) how to combine auxiliary tasks into a single coherent loss. Motived by the issues, this paper proposes a novel Auxiliary Learning frame work, named AuxiLearn. The paper is globally well organized and clearly written.

Pros:
1.	The motivation is straightforward.
2.	The paper proposes sound the technique contributions. Adopting bi-level optimization in Auxiliary Learning makes sense.
3.	The theoretical analysis in this paper supports the efficiency of the method proposed in this paper.
4.	The experimental results and experimental analysis in this paper is plausible.

Cons:
1.	The efficiency of the hypergradient, which is the main shortage, should be discussed.
2.	How to determine the number of iterations J in Alg. 2 is not given.

---

### Author Response · Authors · 2020-11-16
**To All Reviewers**

We thank the reviewers for the insightful feedback. We are encouraged that reviewers found the problem studied significant (R3) and our approach interesting (R4) to a broad audience (R3). Reviewers stated that our paper is clearly written and technically sound (R3) and highlighted the theoretical analysis (R2). Reviewers also appreciated the comprehensive experiments that validated the effectiveness of our approach (R2, R4, R5).

The main concerns were as follows. Reviewers asked for better intuition for learning auxiliaries, and the reasons it works (R2, R3). Reviewers also asked to clarify notations (R2, R4, R5) and the limitations of relying on the IFT  and its approximation (R5). We provide answers to these concerns and describe the changes made to the paper in our responses below.

---

### Decision · Program_Chairs · 2021-01-07
**Final Decision**

**Decision:**

Accept (Poster)

**Comment:**

The paper proposes a novel framework to develop useful auxiliary tasks and combine auxiliary tasks into a single coherent loss. The idea is good and the experiments are sufficient to verify the arguments. All the reviewers agree to accept the paper.